# A comprehensive performance analysis of sequence-based within-sample testing NIPT methods

**Tom Mokveld**[1], **Zaid Al-Ars**[2], **Erik A. Sistermans**[3], **Marcel Reinders**[1]*

**1** Delft Bioinformatics Lab, Delft University of Technology, Delft, The Netherlands, **2** Computer Engineering, Delft University of Technology, Delft, The Netherlands, **3** Department of Human Genetics and Amsterdam Reproduction & Development Research Institute, Amsterdam UMC, Vrije Universiteit Amsterdam, Amsterdam, The Netherlands

* M.J.T.Reinders@tudelft.nl

## Abstract

### Background

Non-Invasive Prenatal Testing is often performed by utilizing read coverage-based profiles obtained from shallow whole genome sequencing to detect fetal copy number variations. Such screening typically operates on a discretized binned representation of the genome, where (ab)normality of bins of a set size is judged relative to a reference panel of healthy samples. In practice such approaches are too costly given that for each tested sample they require the resequencing of the reference panel to avoid technical bias. Within-sample testing methods utilize the observation that bins on one chromosome can be judged relative to the behavior of similarly behaving bins on other chromosomes, allowing the bins of a sample to be compared among themselves, avoiding technical bias.

### Results

We present a comprehensive performance analysis of the within-sample testing method Wisecondor and its variants, using both experimental and simulated data. We introduced alterations to Wisecondor to explicitly address and exploit paired-end sequencing data. Wisecondor was found to yield the most stable results across different bin size scales while producing more robust calls by assigning higher Z-scores at all fetal fraction ranges.

### Conclusions

Our findings show that the most recent available version of Wisecondor performs best.

## Introduction

Non-Invasive Prenatal Testing (NIPT) is designed to detect large genetic abnormalities of the fetus, such as chromosome aneuploidies, sub-chromosomal copy number variations (CNVs), and unbalanced translocations. With the discovery of cell-free fetal DNA (cffDNA) in the

**Data Availability Statement:** The experimental sequencing data cannot be shared publicly because of confidentiality laws. Data are available from the METc VUmc (contact via metc@vumc.nl) for researchers who meet the criteria for access to

confidential data. The discretised simulated sequencing data may be shared.

**Funding:** This work is being funded by the Delft Data Science Center of the Delft University of Technology. The funders had no role in study design, data collection and analysis, decision to publish, or preparation of the manuscript.

**Competing interests:** The authors declare no conflict of interest.

maternal peripheral bloodstream [1], it has become possible to develop NIPT methods to detect genetic abnormalities [2–4]. NIPT offers especially high sensitivity and specificity for common chromosomal aneuploidies such as trisomy 21, 18, and 13 [5, 6], but can also be used for other autosomes [7, 8], and even sub-chromosomal events [9–11]. A critical factor to detect ever smaller CNVs is the depth of coverage, which can become prohibitively costly in practice [12]. Additionally, there are limitations, such as support for only the most common trisomies, and test failure caused by complications such as placental mosaicism or maternal copy number variation [4, 13].

In practice, NIPT methods that utilize whole genome sequencing (WGS) typically rely on extremely low sequencing yield (~0.2x coverage) to remain economical for clinical applications [14, 15]. Besides low coverage there is also the assumption that sufficient cffDNA is present in the maternal plasma. The cffDNA generally contributes only slightly to the complete pool of cfDNA that is sequenced (2–20%), which further complicates accurate identification of CNVs [16, 17]. The amount of cffDNA available in the sample, i.e., the fetal fraction, has a direct bearing on the reliability of any detected CNV, where a higher fetal fraction translates to increased confidence of detected CNVs [18].

Coverage-based NIPT methods typically follow similar steps to detect CNVs [19–23]. DNA is first isolated from maternal plasma and sequenced at very low coverage. The chromosomal origin of each read is then identified through read alignment to the human reference genome. These read alignments are counted and discretized into a coarse representation, binning the genome into equally sized bins. Finally, through statistical testing, the per bin read coverage is compared relative to a reference panel of healthy samples, determining for each bin whether it significantly deviates from the expected signal. This described methodology has a major drawback, being that control samples have to be re-sequenced for each tested sample to avoid technical bias.

An alternative that improves upon these methods is Wisecondor [24], which avoids read frequency variation across different samples. In Wisecondor, each tested bin is compared to a set of reference bins on other chromosomes within the same sample that were found to display similar behavior. Such a within-sample comparison avoids between sample bias and differences within the fetal fraction, since regions with equivalent characteristics will behave similarly within the test sample, and all regions are subject to the same experimental procedures. Wisecondor is freely available and can be modified and improved by the scientific community, as shown with WisecondorX [25], which was built to be a general solution for WGS applications beyond NIPT. One limiting factor of Wisecondor is the utilization of the Stouffer's Z-score sliding window method to segment and score events, which has an exponential computational complexity with respect to a decreasing bin size. Typically, this is not problematic in shallow NIPT, given that only a fraction of the available DNA corresponds to the fetus, necessitating the use of a larger bin size.

Wisecondor was developed during a time when sequencing for NIPT was still utilizing single-end technologies. While Wisecondor can process paired-end data, it does not actually use the additional information that this technology offers. In this study we introduce modifications within Wisecondor to utilize pairing information and include this within a benchmark in which we compare the modified Wisecondor versions to Wisecondor, WisecondorX, and CNVkit [26], an alternative approach to detect CNVs, with both experimental and synthetic data.

## Results

Initially the detection of larger and common aneuploidies was investigated. In total 526 samples were used, of which 401 were confirmed negatives and used as controls while the

remaining 125 were confirmed trisomy 21 positive. The average estimated fetal fraction of all samples is ~7.5%. All samples were aligned using BWA-mem to the hg19 human reference genome with an average depth of coverage of 0.257x across all samples. Wisecondor was modified to utilize full alignment read counting and read pairing. The different versions of Wisecondor are referred to as follows. *WCR*: the baseline implementation of Wisecondor in which no read pairing is available and alignments are counted based on start positions. Three modified implementations of Wisecondor: *WCR+SE*, utilizing the full but unpaired read alignments to determine bin counts; *WCR+PE* and *WCR+PEI* that both use the read pairing information when aligning reads before determining bin counts, and where *WCR+PEI* also contributes to bin counts for the insert size of every paired read. *WCRX*, as WisecondorX. and, lastly, CNVkit was included, which is a general purpose CNV detector, which showed the best competitive performance across different other non-Wisecondor NIPT methods [25].

## T21 detection performance

The most direct measure of performance is the detection rate of expected aberrations in validated samples. Table 1 shows how the methods operate given different bin sizes. The majority of all expected T21s are detected by each method. At the 250 kb scale *WCR+PEI* performs the worst, whereas *WCR* and *WCR+SE* outperform the others, which were found to overlap in all their calls. Note that the originally published version of WCR (which we re-implemented) performs significantly worse than the most recent version, e.g., 118 calls at 250 kb, furthermore the run-time of this legacy implementation was at least an order of magnitude greater, hence the remaining results of this version are omitted. In all but one case *WCR* outperforms the other methods, this being *WCR+PEI* at a 50 kb resolution. The non-Wisecondor method, *CNVkit*, is competitive with the Wisecondor methods, however this performance is expected given the relative ease of detecting whole chromosome events. Given the scale of the events in this context, it is not surprising that the detection rate remains relatively stable as the bin size increases. Performance only regresses significantly at the 50 kb scale, pointing to a failure to segment the events due to the increasing variability within the normalized bins as they become smaller. No significant differences in the run-time of the six methods were found, due to the use of relatively large bin sizes.

## Bin size relates to false positives

While the majority of all expected trisomies were detected by the methods, it is important that the number of false positives remains low. Since discretizing the genome into bins results in a signal/noise trade-off, we summarize the sensitivity across bin sizes for all methods (Fig 1). As expected, a smaller bin size increases the number of false positives that each method detects, except for *WCRX*, this stable performance might be explained by the segmentation algorithm that derives setting breakpoints from the variance across a segment, also resulting in a lower

**Table 1. Detection performance of T21 in experimental data.** The number of $\geq$ 10 Mb events with Z-scores $\geq$ 5 on chromosome 21 detected by the methods in all T21 positive samples for varying bin sizes. Note that *CNVkit* does not depend on bin-sizes so only one performance measure is reported.

|          | 50 kb   | 100 kb  | 250 kb  | 500 kb  | 750 kb  | 1 Mb    | 5 Mb    | 10 Mb   |
|----------|---------|---------|---------|---------|---------|---------|---------|---------|
| **WCR**      | 113     | **122** | **123** | **124** | **123** | **123** | **122** | **121** |
| **WCR+SE**   | 114     | 121     | 122     | 122     | **123** | 121     | 121     | **121** |
| **WCR+PE**   | 117     | 120     | 121     | 121     | 120     | 120     | 120     | 118     |
| **WCR+PEI**  | **118** | 119     | 119     | 119     | 120     | 119     | 118     | 118     |
| **WCRX**     | 78      | 118     | 122     | 122     | 120     | 121     | 120     | 103     |
| **CNVkit**   |         |         |         |  120    |         |         |         |         |

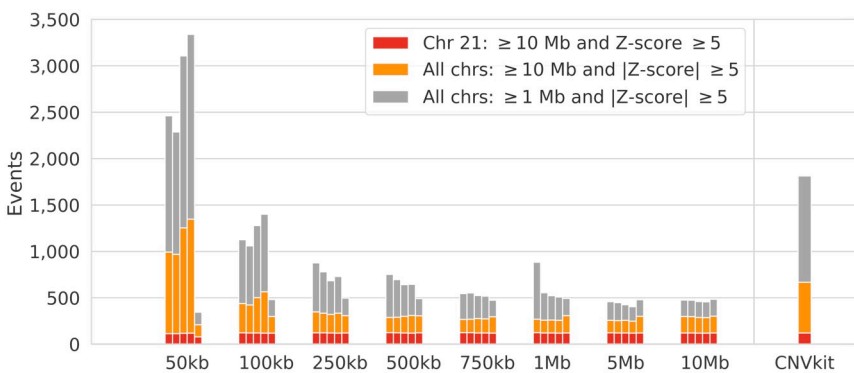

**Fig 1. Bin size relation to detection rate.** Stacked counts of detected CNVs relative to bin size and method, ordered from left to right as: *WCR, WCR+SE, WCR+PE, WCR+PEI, WCRX* (CNVkit results are displayed separately). Bar coloring denotes mutually exclusive filtering constraints. Red: filtering for $\geq 10$ Mb CNVs with Z-score $\geq 5$ (only including duplications) and only on chromosome 21. Orange: filtering for $\geq 10$ Mb CNVs with an absolute Z-score $\geq 5$ (thus also including deletions) and on all chromosomes. Gray: identical to the previous but for all CNVs $\geq 1$ Mb.

detection rate for small bin sizes. The number of false positives for CNVkit is relatively large, giving an advantage for the Wisecondor methods with a larger bin size that have a good control of false positives. Interesting is that *WCR+PE* and *WCR+PEI* become more sensitive compared to *WCR* at lower bin sizes, whereas this relation inverses when the bin size increases. This may be explained by the use of read pairing and insert size padding, which exaggerates large fluctuations in the signal, while smoothing smaller fluctuations. If the bin size becomes smaller it is expected that the signal itself becomes noisier and therefore fluctuate more, the exaggerating effect therefore causes more events to be detected on the lower end. Conversely, the signal becomes increasingly smooth as the bin size increases, which can lead to overly aggressive smoothing, potentially masking true fluctuations.

## Per-bin Z-score differentiation

To understand method-specific differences, the per bin Z-scores were aggregated across all negative and T21 positive samples, as shown in Fig 2. *WCR* and *WCR+SE* perform identically at the 250 kb scale, a similar observation as in Table 1. Furthermore, *WCR+PE*, and especially *WCR+PEI* yield overall lower Z-scores, which can eventually push a potential CNV call below the Z-score threshold. At this scale, the inclusion of read pairing (*WCR+PE*) smooths the read count signal, even more so when also including the insert size padding (*WCR+PEI*). Thus far evidence has shown that this smoothing adversely impacts the detection rate because of overall lower Z-scores at most resolution scales. Yet, an improvement can be noted with *WCR+PE* and *WCR+PEI*, at the cost of many additional false positives, when the bin size becomes smaller, being the best performing at a 50 kb scale (Table 1). In fact, the average per bin Z-scores at a 50 kb scale does show that the scores of these two methods are overall higher than that of the others (S1 Fig).

## Detection power relative to fetal fraction

To further understand the differences in Z-scores between the methods, the Z-scores of individual events were considered in relation to the estimated fetal fraction of each sample. The expectation is that samples with higher fetal fractions can be more reliably tested for CNVs. This also implies that the Z-scores of events are likely higher for calls made in samples with higher fetal fractions. Fig 3 shows the Z-scores of the events on chromosome 21 for the T21

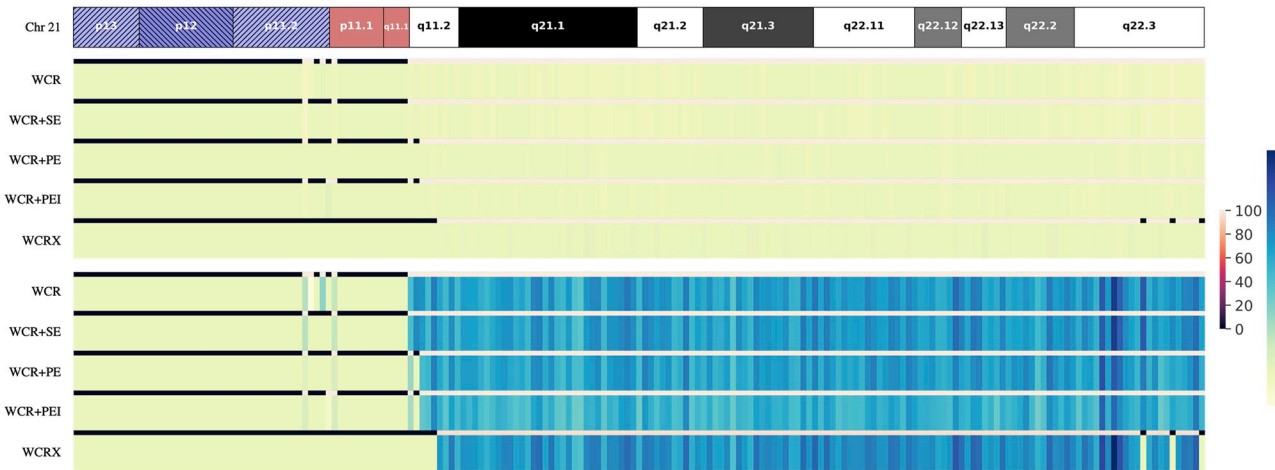

**Fig 2. Per-bin Z-score relationship at 250 kb.** Heatmaps of the summed per bin Z-scores across all negative (top) and all T21 positive (bottom) samples at a 250 kb bin scale for chromosome 21 and all different Wisecondor-based methods. The line above each method's heatmap corresponds to the average number of selected reference bins for each bin of that method (black denoting that no similar reference bins are found and consequently these bins are excluded).

positive samples with respect to the estimated fetal fractions (S2 Fig for other bin sizes). Indeed, there is a positive correlation of the fetal fraction with respect to the Z-score of the detected events for all methods. It also shows, again, that, at a bin size of 250 kb, *WCR* assigns higher Z-scores to detected events than any other method. With exception of *WCR+SE* that performs nearly identical, as can be observed from the near exact same overlapping predictions and regression in Fig 3. This shows that *WCR* has greater power to detect expected T21s, and thus detects events with more confidence.

## Performance in simulated data

As there is no true ground truth available in the experimental data, performance could not be measured in concrete terms, such as on the breakpoints of events. Therefore, synthetic read data was created. A total of 400 positive samples were generated, equally split for duplications, deletions (both at varying scales), and sex (*WCRX* generates sex-specific reference sets) with a constant fetal fraction set at 7.5%. Additionally, 100 negative samples were generated, which were used to build reference sets. With the ground truth known, the TP, TN, FP, and FN rates can be determined directly from the intervals of the simulated events, the detected events, and the chromosomes. Fig 4 shows an overview of the F1 performance of the methods for different simulated events and event sizes. Table 2 shows the average F1 of duplications and deletions across all event scales and false positives calls. The overall performance of all methods is similar, with *WCR* performing best. Peculiar to *WCRX* is that the performance is poor in samples with 1 Mb events while stabilizing from 5 Mb and up until regressing again as the event size increases from 30 Mb and beyond. It is unclear what causes this behavior. However, in terms of false positive calls on other chromosomes (Table 2), *WCRX* does best with the fewest additional calls across all samples. *CNVkit*, increasingly struggles as the event size becomes smaller, calling many false positives.

## Performance in challenging simulated data

The previous results highlight that, generally, events larger than 5 Mb can be detected by all methods, which signals that the real challenge lies within events that are smaller than this.

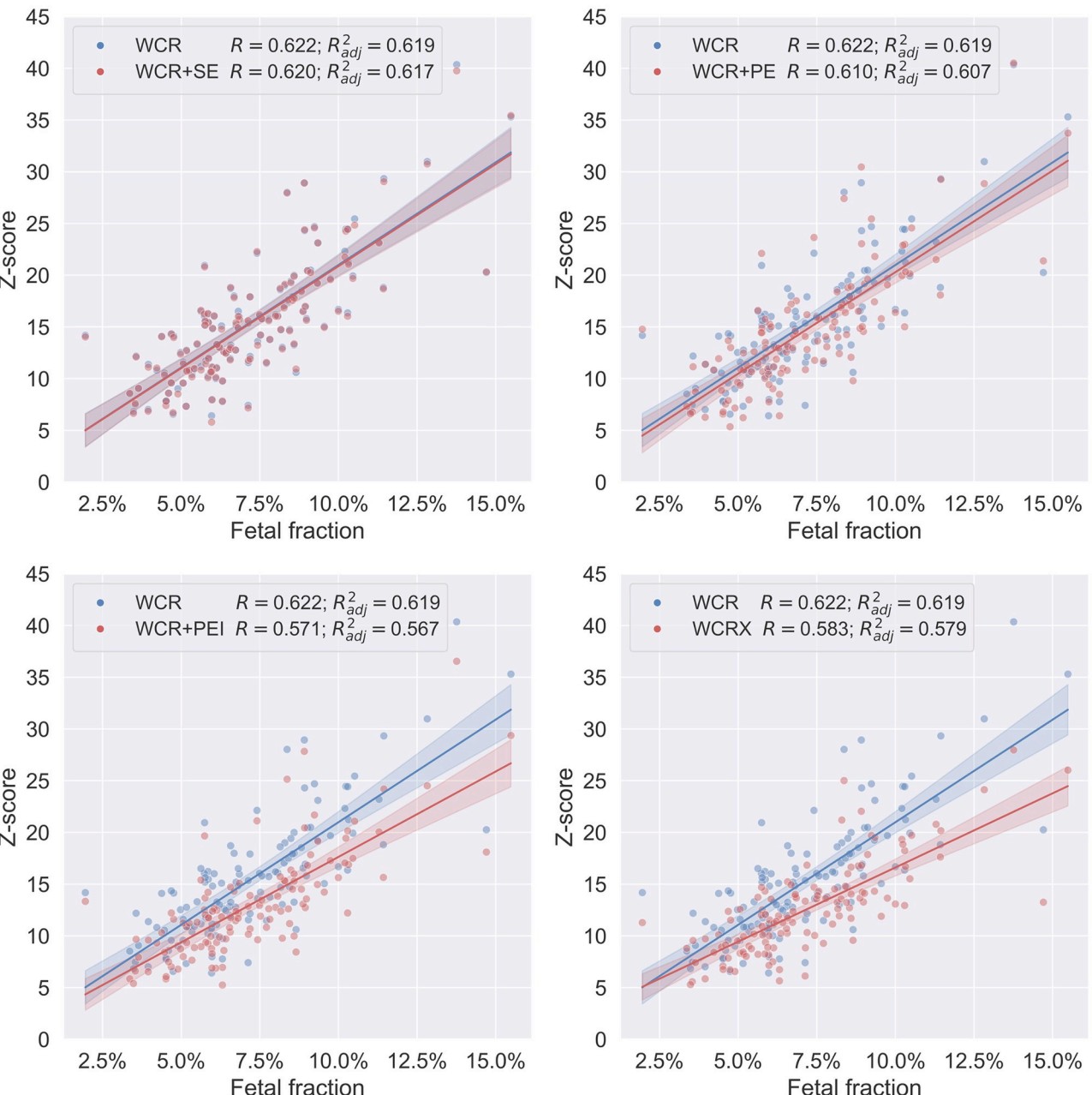

**Fig 3. Event Z-scores relative to fetal fraction at 250 kb.** All $\geq$ 10 Mb events with Z-scores $\geq$ 5 on chromosome 21, detected by the different methods for a 250 kb bin size in the T21 positive samples, relative to the estimated fetal fractions of each sample. Each plot compares one of the methods WCR +SE, *WCR+PE*, *WCR+PEI*, and *WCRX* (all in red) with *WCR* (in blue), and each point corresponds to an event within a sample. The legend annotation relates each method with the respective R value and adjusted R squared value of the linear fit.

Those events are also on the edge of what remains detectable with respect to the samples' read coverage and fetal fraction. To further investigate this, 480 additional simulated samples were generated, now with aberrations ranging from 250 kb to 5 Mb and fetal fractions ranging between 1–6%. Methods were again evaluated for a 250 kb bin size, and the results are shown in Fig 5 for a fetal fraction of 6%, and the accumulated results for all fetal fraction ranges in Table 3.

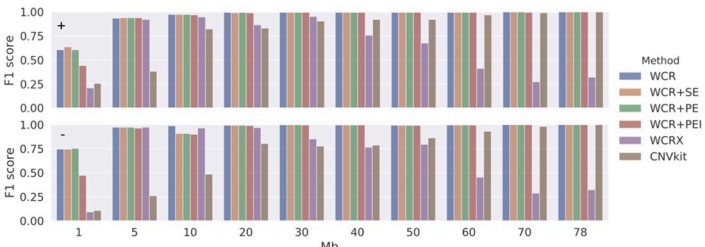

**Fig 4. F1 performance in simulated data.** F1 score (vertical axes) for simulated duplication (+, top panel) and deletion (-, bottom panel) events of varying sizes (1 to 78 Mb) at a fixed fetal fraction of 7.5% for the different methods using a 250 kb bin size.

**Table 2. Averaged F1 performance in simulated data.** F1 score as averaged across all event sizes (1 to 78 Mb) for simulated duplications (+), deletions (-), and the number of false positive calls (FP) of the different methods using a 250 kb bin size.

|  | F1 (+) | F1 (-) | FP |
|---|---|---|---|
| **WCR** | **0.996** | **0.995** | 33 |
| **WCR+SE** | 0.995 | **0.995** | 32 |
| **WCR+PE** | 0.995 | **0.995** | 30 |
| **WCR+PEI** | 0.994 | 0.986 | 40 |
| **WCRX** | 0.577 | 0.605 | **2** |
| **CNVkit** | 0.929 | 0.845 | 556 |

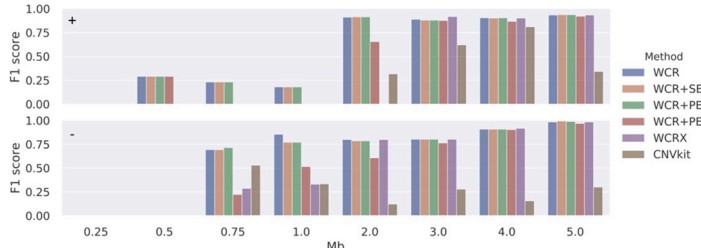

**Fig 5. F1 performance in challenging simulated data.** F1 score for simulated duplication (+, top panel) and deletion (-, bottom panel) events of varying sizes (0.25 to 5 Mb) at a fixed fetal fraction of 6% for the different methods using a 250 kb bin size.

As events become smaller than 1 Mb, the likelihood of detecting an event becomes extremely small under the given conditions. As expected, an increasing fetal fraction improves performance of every method. As the fetal fraction increases (>4%) *WCRX* performs closely to the other Wisecondor methods but does worse otherwise. Note that *WCRX* does not exhibit the dramatic drop in performance at lower event scales as was observed when events became larger than 20 Mb (Fig 4). The performance of *CNVkit* is again significantly worse than the Wisecondor methods when events become increasingly smaller, even more so for deletions, which is not the case for the other methods which are balanced for both types of events. When utilizing the insert size padding of *WCR+PEI*, the performance for these smaller events is always worse than with the other Wisecondor methods. This result is consistent with the previous findings when varying the bin size. In terms of false positive calls on other chromosomes across all samples, *WCRX* again does best with the fewest additional calls (Table 3).

**Table 3. Averaged F1 performance in challenging simulated data.** F1 score as averaged across all event sizes (0.25 to 5 Mb) for simulated duplication (+) and deletion (-) events across varying fetal fractions (FF), and the number of false positive (FP) calls of the different methods at a 250 kb bin size.

| | FF | WCR | WCR+SE | WCR+PE | WCR+PEI | WCRX | CNVkit |
|---|---|---|---|---|---|---|---|
| + | 1% | 0 | 0 | 0 | 0 | 0 | 0 |
| | 2% | **0.108** | **0.108** | **0.108** | 0 | 0 | 0 |
| | 3% | 0.598 | 0.572 | **0.609** | 0.379 | 0.327 | 0.114 |
| | 4% | 0.759 | **0.763** | 0.760 | 0.511 | 0.536 | 0.237 |
| | 5% | 0.839 | 0.838 | **0.840** | 0.775 | 0.770 | 0.408 |
| | 6% | **0.837** | 0.836 | **0.837** | 0.794 | 0.769 | 0.474 |
| - | 1% | 0 | 0 | 0 | 0 | 0 | 0 |
| | 2% | 0.227 | **0.297** | 0.247 | 0.090 | 0.156 | 0.125 |
| | 3% | 0.522 | **0.525** | 0.524 | 0.295 | 0.380 | 0.155 |
| | 4% | 0.665 | **0.715** | 0.710 | 0.658 | 0.696 | 0.157 |
| | 5% | **0.845** | 0.842 | 0.842 | 0.781 | 0.808 | 0.190 |
| | 6% | **0.863** | 0.862 | 0.862 | 0.799 | 0.835 | 0.210 |
| | FP | 125 | 117 | 126 | 134 | 1 | 515 |

## Discussion

CNV detection with highly imbalanced mixed samples remains a challenging problem. Within the low yield NIPT context this challenge is further exacerbated given that the lines become increasingly blurred between the fetal, maternal and noise signals. This happens to such an extent that resolution must be sacrificed to increase the signal-to-noise ratio sufficiently to allow for confident calls to be made. This is even more the case at lower fetal fractions. Generally, this compromise happens by aggregating the read count signal across a larger range, discretizing the genome into bins. The chosen bin size can thus be understood as a direct trade-off between resolution, noise, read coverage and fetal fraction notwithstanding.

Methods that utilize within-sample correction such as Wisecondor, offer a solution to the issue of having to re-sequence control samples for each set of new samples that must be tested. We have shown, in multiple instances, that for NIPT applications, overall, Wisecondor (*WCR*) performs best relative to the other methods that were tested, in the experimental as well as the synthetic data, both in terms of better performance and by assigning higher Z-scores to any detected CNVs. In general, all Wisecondor methods performed better than the non-Wisecondor method, CNVkit, especially when considering lower fetal fraction and smaller CNVs. This highlights the importance of tailoring methods to specific domains, regardless of the similarity of the underlying problem, detecting CNVs. A notable advantage of WisecondorX is that this method detects far fewer spurious findings, i.e., the CNVs that are called are much more likely to be actual events, while this is at the expense of a lower detection rate. In the NIPT setting a lower number of false positives is beneficial given that all findings are typically scrutinized, meaning that WisecondorX would allow for a more efficient workflow.

Several modifications to Wisecondor were introduced. First, rather than counting the start position of each read the full alignments are processed instead (*WCR+SE*). We noted that it is not as beneficial as expected in this setting given the sparseness of the read data, resulting in nearly identical results to Wisecondor (*WCR*). However, such processing may become more useful when coverage increases and the bin size is allowed to become smaller, as the effect of this change would then be amplified. Since Wisecondor does not utilize the pairing between reads, a modification was added that exploits paired-end reads (*WCR+PE*), as well as an adaptation that performs additional padding using the insert size obtained from

these reads (*WCR+PEI*). Our experiments showed that these modified versions allow for higher sensitivity at smaller bin sizes by exaggerating fluctuations within the coverage signal, but that this is at the cost of overly smoothing the signal at larger bin sizes, losing sensitivity. While smoothing the signal was beneficial, this was only so when bin sizes became increasingly smaller, at the costs of increased false positive calls.

The contribution of utilizing read pairing in this study had only a modest effect on the performance of the methods. To the point that using paired-end reads as single-end reads can perform better. However, there is more information within the paired-end reads that was not utilized. Possible improvements for any method utilizing paired-end reads can exploit the fact that fetal DNA fragments have a shorter fragment size than maternal fragments [27]. The characteristic of fetal DNA fragments having shorter fragment size than maternal fragments is a result of the underlying mechanisms involved in DNA fragmentation, such as DNA methylation and its relation to chromatin accessibility [28, 29]. The fragment size differences can be inferred from the insert size of the aligned paired-end reads. Several methods have been proposed that leverage this fragment size difference to detect large chromosomal CNVs [30] or fetal de novo point mutations [31]. Furthermore, combining the fragment size difference with the read count signal has enabled the improved detection of fetal aneuploidies, as demonstrated by the COF-FEE algorithm [32], which is a reference free method requiring no control samples. Lastly, WisecondorFF [33] was developed by our group to extend the WISECONDOR within sample testing framework and facilitate the combination of read count and fragment sizes to improve (sub) chromosomal CNV calling. We also note that method performance is significantly impacted by the choice of bin size, such that it may be beneficial to utilize multiple bin scales, making it possible that a CNV may gain support across multiple bin scales rather than only one.

## Methods

### Sample specification and pre-processing

All samples are derived from the Dutch TRIDENT study [34]. The study was conducted according to the guidelines of the Declaration of Helsinki and approved by the Institutional Review Board (or Ethics Committee) of VU University Medical Center Amsterdam (protocol code 2021.0515 –DISTIL, date of approval 21-09-2021). Written informed consent was obtained from all subjects involved in the study. DNA isolation, library preparation and paired-end sequencing (36bp) were performed using the Illumina VeriSeq1 sequencing protocol, according to the recommendations of the supplier (Illumina, San Diego, USA). Analysis was performed by both the Veriseq algorithm (which only detects trisomies 21, 13 and 18), and by Wisecondor, which also detects other trisomies and smaller events. For this study 526 samples were selected, of which 401 had no detected chromosomal aberrations and were used as negative controls. The remaining 125 samples all tested positive for T21. All samples were similarly preprocessed, and aligned to the hg19 human reference genome, excluding any decoy sequences, using BWA-0.7.17 mem [35]. Since both single-end and paired-end performance is measured, the read sets were aligned in both settings. Meaning that in the single-end setting each read pair was individually aligned before merging the output alignments. This as opposed to the paired-end setting in which all alignments were output at once. The average coverage of the 401 negatives is 0.258, and that of the 125 T21 positives is 0.256. SeqFF was utilized to estimate the fetal fractions of all samples, with the average fetal fraction being 7.5% [36]. The final alignments were compressed and left unfiltered, given that all methods internally control the quality of the alignments.

For each of the samples, initial genome wide bin counts were generated at a 5 kb resolution, these counts were scaled up to 50 kb, 100 kb, 250 kb, 500 kb, 750 kb, 1 Mb, 5 Mb, and 10 Mb

for the construction of the reference panels and/or the testing of a sample against these references. All 401 control samples were used to build reference panels at every bin scale for each of the 5 methods: *WCR*, *WCR+SE*, *WCR+PE*, *WCR+PEI*, and *WCRX*. No predefined blacklist was used to exclude genomic regions from the analysis, instead the methods define those based on the normalized bin counts.

## Wisecondor modification

We introduced three different modifications to Wisecondor (version commit: 9e95c75), note that this is not the published version of the algorithm, but an updated variant. One that exploits the full read alignment instead of the start position (*WCR+SE*), and two that exploit the pairing between reads, one requiring proper pairing of reads (*WCR+PE*) and one additionally utilizing the fragment size associated with the paired reads (*WCR+PEI*). The genome wide bin counts are unique for each Wisecondor variant, such that these counts must be generated for each.

*WCR+SE*: The method of counting aligned reads in the discretized bins was modified and is based on the full alignments, rather than the starting positions of the aligned reads. Doing so the full information content of an alignment can be utilized, as well as allow reads to partially contribute to multiple bins. Every modified variant of Wisecondor, denoted as *WCR +*, uses this alternate read counting.

*WCR+PE*: Wisecondor does not utilize paired-end read information but treats any aligned read as single-end. By requiring any read to be in a properly paired pair, an additional constraint is imposed for a read to be considered.

*WCR+PEI*: Additionally, the insert size distance between two properly paired fragments can be utilized to further pad the read coverage such that the fragment between the aligned read pairs also contributes to the total contribution of the reads. By using read pairing, the read count signal is smoothed and large fluctuations within the signal are exaggerated. Such signal smoothing and exaggeration is especially prominent in *WCR+PEI* (S3 Fig).

## Data simulation

Chromosome 18 was selected to simulate duplications and deletions of size: 1, 5, 10, 20, 30, 40, 50, 60, 70, and 78 Mb. For every CNV size, 10 samples were simulated for each sex and variation type, resulting in a total of 400 samples. Additionally, 100 negative samples were simulated, 50 of each sex. To generate an aberrated sample, a random starting position was selected for the event on chromosome 18, excluding highly repetitious regions, using a uniform distribution. Given the starting position up to the end position, the sequence is either deleted, leaving flanking segments of N's (larger than the fragment size of the simulated reads) at the site, or duplicated, leaving flanking segments of N's between the two sequences. For each of the aberrated samples, full fetal reference sequences were created by replacing the original chromosome 18 sequence with that of the modified sequence. The sex further determined the inclusion/exclusion of Y or one X chromosome.

For each of the samples a total of 21 million 36 bp paired-end reads were simulated using Mason 2.0.9 [37]. The read sets were generated as such, that a fetal fraction of 7.5% would be observed when merged. Given that the fetal fragment length is shorter than the maternal fragments, the mean fragment length parameter of fetal samples was reduced to be slightly shorter than that of the maternal fragments. A maternal read set (reads prefixed with M and a total of 19,425,000 reads) was then simulated, using the hg19 reference sequence without chromosome

Y and decoy sequences, and a fetal read set (reads prefixed with F and a total of 1,575,000 reads) from the previously generated fetal reference sequences. The two disjoint paired-end read sets are finally merged and prefix sorted to eliminate any aligner bias.

## Performance metrics

Within the simulated data the ground truth of a CNV, $T$, can be represented as an interval, denoted as $[T_s, T_e]$, which may be contained within a chromosome, $G$, i.e., a larger interval, as $[G_s, G_e]$. For a CNV predicted by one of the methods, $P$, the same can be done, as the interval $[P_s, P_e]$, or the lack thereof as $[P_s = 0, P_e = 0]$. The true positives, can then be derived to be $TP = \max(0, \min(T_e, P_e))$; the false negatives, as $FN = T_e - T_s - TP$; the false positives, as $FN = P_e - P_e - TP$; the true negatives, as $TN = G_e - TP - FN - FP$. The $F_1$ performance score, i.e., the harmonic mean of the precision and recall, may then be calculated as: $F_1 = \frac{TP}{TP + \frac{1}{2}(FP + FN)}$.

## Supporting information

**S1 Fig. Per-bin Z-score relationship at 50 kb.** Heatmap of the summed per bin Z-scores across all negative (top) and all T21 positive (bottom) samples at a 50 kb bin scale for chromosome 21 and all different Wisecondor-based methods. The line above each method's heatmap corresponds to the average number of selected reference bins for each bin of that method (black denoting that no similar reference bins are found and consequently these bins are excluded). (TIF)

**S2 Fig. Event Z-scores relative to fetal fraction at varying bin size.** All $\geq 10$ Mb events with Z-scores $\geq 5$ on chromosome 21 detected by the different methods in the 125 T21 positive samples relative to the estimated fetal fractions of each sample. Each plot displays one of the methods WCR+SE, WCR+PE, WCR+PEI, and WCRX (all in red) overlaid with WCR (shown in blue), for a set bin-size resolution. Each point corresponds to a CNV within a sample. (TIF)

**S3 Fig. Effects of Wisecondor modifications on simulated data.** Simulated read alignment counts across a small genome of single-end, paired-end, and paired-end with insert padding methods. On top this is shown for an unaffected sample, and on the bottom for a sample with a 400 bp deletion. (TIF)

## Acknowledgments

We would like to thank Daoud Sie for his technical assistance and his efforts in making the experimental data available.

## Author Contributions

**Conceptualization:** Tom Mokveld, Marcel Reinders.

**Data curation:** Tom Mokveld, Erik A. Sistermans.

**Formal analysis:** Tom Mokveld.

**Investigation:** Tom Mokveld.

**Methodology:** Tom Mokveld.

**Project administration:** Erik A. Sistermans, Marcel Reinders.

**Resources:** Erik A. Sistermans.

**Software:** Tom Mokveld.

**Supervision:** Marcel Reinders.

**Validation:** Zaid Al-Ars, Erik A. Sistermans, Marcel Reinders.

**Visualization:** Tom Mokveld.

**Writing – original draft:** Tom Mokveld.

**Writing – review & editing:** Tom Mokveld, Zaid Al-Ars, Erik A. Sistermans, Marcel Reinders.

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
