## [Decision Letter · Decision Letter 0]

12 Jan 2023

PONE-D-22-24416A comprehensive performance analysis of sequence-based within-sample testing NIPT methodsPLOS ONE

Dear Dr. Reinders,

Thank you for submitting your manuscript to PLOS ONE. After careful consideration, we feel that it has merit but does not fully meet PLOS ONE’s publication criteria as it currently stands. Therefore, we invite you to submit a revised version of the manuscript that addresses the points raised during the review process. Please reply to the reviewers' comments and submit your revised manuscript by Feb 26 2023 11:59PM. If you will need more time than this to complete your revisions, please reply to this message or contact the journal office at plosone@plos.org. Please include the following items when submitting your revised manuscript:A rebuttal letter that responds to each point raised by the academic editor and reviewer(s). You should upload this letter as a separate file labeled 'Response to Reviewers'.A marked-up copy of your manuscript that highlights changes made to the original version. You should upload this as a separate file labeled 'Revised Manuscript with Track Changes'.An unmarked version of your revised paper without tracked changes. You should upload this as a separate file labeled 'Manuscript'.If applicable, we recommend that you deposit your laboratory protocols in protocols.io to enhance the reproducibility of your results. Protocols.io assigns your protocol its own identifier (DOI) so that it can be cited independently in the future. For instructions see: https://journals.plos.org/plosone/s/submission-guidelines#loc-laboratory-protocols. Additionally, PLOS ONE offers an option for publishing peer-reviewed Lab Protocol articles, which describe protocols hosted on protocols.io. Read more information on sharing protocols at https://plos.org/protocols?utm_medium=editorial-email&utm_source=authorletters&utm_campaign=protocols.

We look forward to receiving your revised manuscript.

Kind regards,

Hao Sun, Ph.D.

Academic Editor

PLOS ONE

Journal Requirements:

This work is being funded by the Delft Data Science Center of the Delft University of Technology, which has no role in the design and execution of the study as well as the interpretation of the data and writing of the manuscript

Additional Editor Comments:

The manuscript is technically sound but need to be substantially revised before publication based on reviewers' comments.

Reviewers' comments:

Reviewer's Responses to Questions

**Comments to the Author**

1. Is the manuscript technically sound, and do the data support the conclusions?

Reviewer #1: Yes

Reviewer #2: Yes

2. Has the statistical analysis been performed appropriately and rigorously? 

Reviewer #1: N/A

Reviewer #2: Yes

3. Have the authors made all data underlying the findings in their manuscript fully available?

Reviewer #1: No

Reviewer #2: Yes

4. Is the manuscript presented in an intelligible fashion and written in standard English?

Reviewer #1: Yes

Reviewer #2: Yes

5. Review Comments to the Author

Reviewer #1: The manuscript by Mokveld et al. reported the performance of WISECONDOR and its variants on CNV detection using real and simulated data. The manuscript is technically sound, but need to be substantially revised before publication.

The abstract and introduction talks about CNVs while the Results are mostly about Trisomy 21 test. The authors should be clear what is the focus of this work? For prenatal testing, Trisomy 21 is the most common case therefore focusing on it is acceptable.

COFFEE algorithm (Sun et al. Prenat Diagn. 2017, PMID: 28165140) is another control-free implementation for Trisomy 21 test. The authors need to discuss the differences between WISECONDOR and COFFEE.

For PE data, do the authors need to re-mine the control regions for WISECONDOR-PE? Since fetal DNA is known to be shorter, and utilizing such information could help to improve the performance of Trisomy 21 test (Sun et al. PNAS 2018, PMID: 29760053; Chan et al. PNAS 2016, PMID: 27799561), but I did not see such implementation in WISECONDOR-PE? The authors need to discuss it.

For Table 1, what are the specificities? Did the authors use all the control samples to build the model therefore could not calculate a specificity?

For batch-to-batch variation, citations are needed. For instance, Sun et al. Prenat Diagn. 2017 (PMID: 28165140).

The reference list is incomplete. I saw [34, 35] in line 275, but I could not find them in Reference. Also I did not see 31-33 in the main text.

Reviewer #2: The paper “A comprehensive performance analysis of sequence-based within sample testing NIPT methods” compared the performance of the within-sample testing method Wisecondor and its variants, using both experimental and simulated data. The author concluded Wisecondor yielded the most stable results across different bin size scales while producing more robust calls by assigning higher Z-scores at all fetal fraction ranges. However, the scientific novelty of the paper is quite limited, and the paper does not bring significant new insights to the NIPT field.

1) Line 83-85, “All samples were aligned using BWA-mem to the hg19 human reference genome with an average depth of coverage of 0.257x across all samples.” The newest released version of human reference genome is hg38. And hg38 is an improved version of hg19 with higher precision. It would be better to use newer assembly for alignment in this paper.

2) Line 166-167, “Indeed, there is a positive correlation of the fetal fraction with respect to the Z-score of the detected events for all methods.” Figure 3 showed a positive trend, but the correlation coefficient (measure the strength of a liner relationship) between the fetal fraction with the related Z-score was not described.

3) In Figure 3, the upper two figures showed red lines while blue lines were invisible. In this case, it will be helpful if the author can state whether WCR+SE and WCR+PE (all in red) overlaid with WCR (shown in blue).

4) In this paper, the author mainly compared the performance of T21 events. A paper published by your group “WISECONDOR: detection of fetal aberrations from shallow sequencing maternal plasma based on a within-sample comparison scheme” have presented Chromosome 19 was a difficult chromosome for WISECONDOR because of CG richness at Chromosome 19. It would be more comprehensive if other Chromosome comparisons among Wisecondor and its variants such as Chromosome 19 were included.

5) In addition to the comparisons shown in the paper, one of the crucial issues for evaluating programs is runtime. Is there a significant difference in computing times between Wisecondor with its variants?

6. PLOS authors have the option to publish the peer review history of their article (what does this mean?). If published, this will include your full peer review and any attached files.

Reviewer #1: No

Reviewer #2: No

---

## [Author Response · Author response to Decision Letter 0]

1 Feb 2023

Responses to reviewer and editor comments may be found under "Response to Reviewers".

---

## [Decision Letter · Decision Letter 1]

22 Mar 2023

PONE-D-22-24416R1A comprehensive performance analysis of sequence-based within-sample testing NIPT methodsPLOS ONE

Dear Dr. Reinders,

Thank you for submitting your manuscript to PLOS ONE. After careful consideration, we feel that it has merit but does not fully meet PLOS ONE’s publication criteria as it currently stands. Therefore, we invite you to submit a revised version of the manuscript that addresses the points raised during the review process.

We look forward to receiving your revised manuscript.

Kind regards,

Hao Sun, Ph.D.

Academic Editor

PLOS ONE

Journal Requirements:

Additional Editor Comments (if provided):

One of the reviewers has raised some addition questions list below. Kindly revise the manuscript according to the comments from the reviewers.

Reviewers' comments:

Reviewer's Responses to Questions

**Comments to the Author**

1. If the authors have adequately addressed your comments raised in a previous round of review and you feel that this manuscript is now acceptable for publication, you may indicate that here to bypass the “Comments to the Author” section, enter your conflict of interest statement in the “Confidential to Editor” section, and submit your "Accept" recommendation.

Reviewer #1: All comments have been addressed

Reviewer #2: All comments have been addressed

2. Is the manuscript technically sound, and do the data support the conclusions?

Reviewer #1: Yes

Reviewer #2: Yes

3. Has the statistical analysis been performed appropriately and rigorously? 

Reviewer #1: Yes

Reviewer #2: Yes

4. Have the authors made all data underlying the findings in their manuscript fully available?

Reviewer #1: Yes

Reviewer #2: Yes

5. Is the manuscript presented in an intelligible fashion and written in standard English?

Reviewer #1: Yes

Reviewer #2: Yes

6. Review Comments to the Author

Reviewer #1: The authors had addressed most of my comments. However, I found that the citations are still inaccurate, especially the COFFFEE paper (ref. 34): the author list is completely wrong. In addition, the author had discussed and utilized fragmentation of fetal-dervied cfDNA in NIPT, then they should also cite Sun et al. PNAS 2018 (PMID: 29760053) and a recent paper An et al. Nature Communications 2023 (PMID: 36653380) as these papers reveals the mechanisms to fetal DNA fragmentation for improving NIPT. The authors must take the citation seriously and correct all the errors.

Reviewer #2: The author have addressed all my comments. They explained why they focus on chromosome 18 and 21 instead of investigate all chromosomes.

7. PLOS authors have the option to publish the peer review history of their article (what does this mean?). If published, this will include your full peer review and any attached files.

Reviewer #1: No

Reviewer #2: No

---

## [Author Response · Author response to Decision Letter 1]

26 Mar 2023

Reviewer 1:

R1.1. The authors had addressed most of my comments. However, I found that the citations are still inaccurate, especially the COFFFEE paper (ref. 34): the author list is completely wrong. In addition, the author had discussed and utilized fragmentation of fetal-derived cfDNA in NIPT, then they should also cite Sun et al. PNAS 2018 (PMID: 29760053) and a recent paper An et al. Nature Communications 2023 (PMID: 36653380) as these papers reveals the mechanisms to fetal DNA fragmentation for improving NIPT. The authors must take the citation seriously and correct all the errors.

We thank the reviewer for their attention to detail and apologize for the inaccuracies in our citations. We have addressed this issue and taken measures to correct this oversight. Furthermore, we have carefully examined all other citations to ensure their accuracy and completeness. 

The reviewer raises a fair point that along with noting the difference in fragment size between fetal and maternal fragments, it is essential to include citations that investigate the underlying mechanisms responsible for this observed distinction.

Changes in manuscript:

Citations have been corrected and renumbered

The last paragraph has been extended to include the new citations with a short justification: "The characteristic of fetal DNA fragments having shorter fragment size than maternal fragments is a result of the underlying mechanisms involved in DNA fragmentation, such as DNA methylation and its relation to chromatin accessibility [28,29]. The fragment size differences can be inferred from the insert size of the aligned paired-end reads."

---

## [Editor Report · Decision Letter 2]

3 Apr 2023

A comprehensive performance analysis of sequence-based within-sample testing NIPT methods

PONE-D-22-24416R2

Dear Dr. Reinders,

We’re pleased to inform you that your manuscript has been judged scientifically suitable for publication and will be formally accepted for publication once it meets all outstanding technical requirements.

Kind regards,

Hao Sun, Ph.D.

Academic Editor

PLOS ONE
---

## [Editor Report · Acceptance letter]

5 Apr 2023

PONE-D-22-24416R2 

A comprehensive performance analysis of sequence-based within-sample testing NIPT methods 

Dear Dr. Reinders:

I'm pleased to inform you that your manuscript has been deemed suitable for publication in PLOS ONE. Congratulations! Your manuscript is now with our production department. 

Kind regards, 

on behalf of

Dr. Hao Sun 

Academic Editor

PLOS ONE